# The debranching enzyme Dbr1 regulates lariat turnover and intron splicing

Luke Buerer [1], Nathaniel E. Clark [1], Anastasia Welch[1], Chaorui Duan[1], Allison J. Taggart[1], Brittany A. Townley[2], Jing Wang[1], Rachel Soemedi[1], Stephen Rong [3,4], Chien-Ling Lin [1,5], Yi Zeng[6,7], Adam Katolik[8], Jonathan P. Staley [6], Masad J. Damha [8], Nima Mosammaparast [2] & William G. Fairbrother[1,3] ✉

The majority of genic transcription is intronic. Introns are removed by splicing as branched lariat RNAs which require rapid recycling. The branch site is recognized during splicing catalysis and later debranched by Dbr1 in the rate-limiting step of lariat turnover. Through generation of a viable *DBR1* knockout cell line, we find the predominantly nuclear Dbr1 enzyme to encode the sole debranching activity in human cells. Dbr1 preferentially debranches substrates that contain canonical U2 binding motifs, suggesting that branchsites discovered through sequencing do not necessarily represent those favored by the spliceosome. We find that Dbr1 also exhibits specificity for particular 5′ splice site sequences. We identify Dbr1 interactors through co-immunoprecipitation mass spectrometry. We present a mechanistic model for Dbr1 recruitment to the branchpoint through the intron-binding protein AQR. In addition to a 20-fold increase in lariats, Dbr1 depletion increases exon skipping. Using ADAR fusions to timestamp lariats, we demonstrate a defect in spliceosome recycling. In the absence of Dbr1, spliceosomal components remain associated with the lariat for a longer period of time. As splicing is co-transcriptional, slower recycling increases the likelihood that downstream exons will be available for exon skipping.

The removal of introns from pre-mRNA by splicing is an integral step in gene expression. Splicing is a two-step process performed by the spliceosome, a large macromolecular machine that rivals the ribosome in complexity. The first step involves the generation of a 2′-5′ linkage between the 5′ splice site and a branchpoint nucleotide that typically resides near the 3′ end of the intron. In the second step, the free 5′ exon then attacks the 3′ splice site of the downstream exon, leading to exon ligation and the release of the intron lariat. Both the 5′ splice site cleavage and the exon ligation steps of splicing have been shown to be reversible[1], and kinetic proofreading processes mediate the correct selection of splice sites and branchpoints[2]. Splicing assays on mutated branchsites demonstrate that, while the canonical YTNAY branchsite motif is most efficient, the spliceosome can process substrates with flexibility in both branchpoint bulge position and nucleotide identity[3]. Adenosine (A) branchpoints are most efficiently spliced while G branchpoints exhibit a lower

[1]Department of Molecular Biology, Cell Biology, and Biochemistry, Brown University, Providence, RI 02903, USA. [2]Department of Pathology & Immunology, Center for Genome Integrity, Washington University School of Medicine, St. Louis, MO 63110, USA. [3]Center for Computational Molecular Biology, Brown University, Providence, RI 02912, USA. [4]Department of Genetics, Yale University, New Haven, CT 06520, USA. [5]Institute of Molecular Biology, Academia Sinica, Taipei 115, Taiwan. [6]Department of Molecular Genetics and Cell Biology, University of Chicago, Chicago, IL 60637, USA. [7]Department of Genetics, Stanford University School of Medicine, Stanford, CA 94305, USA. [8]Department of Chemistry, McGill University, Montreal, QC H3A 0B8, Canada. ✉e-mail: william_fairbrother@brown.edu

splicing activity and U and C branchpoints have the lowest abilities to support splicing.

First identified as a modifier of retrotransposon activity in yeast, the lariat-debranching enzyme Dbr1 is responsible for the linearization of intron lariats – the rate limiting step in intron turnover[4,5]. The alignment of high-throughput RNA sequencing reads to the human genome has mapped core gene expression elements like start, splice and polyadenylation sites to saturation in many tissues[6]. The branchpoint is the only obligate cis-element of intron-containing genes that has not been mapped to a similar level of completion.

The current set of annotated branchpoints has been discovered by lariat sequencing, a technique that exploits the ability of reverse transcriptase to read through a branchpoint while copying a lariat into cDNA[7]. Our lab created lariat-seq by adapting this method to high throughput sequencing data[8]. These RNA-seq reads cannot be mapped with traditional aligners, but unique mapping methods have been developed to capture the valuable information they provide on branchpoint usage and overall lariat levels While subsequent implementations restricted the search window and added filtering, all share the original approach of aligning reads in a gapped, inverted fashion to the genome producing two alignment blocks: an upstream block adjacent to the 5′ splice site and a downstream block in the 3′ end of the intron where the point of discontinuity corresponds to the branchpoint[8–10]. Another method to detect the signature of lariats in RNA-Seq, Shapeshifter, utilized the shape of RNA-seq coverage curves across the boundary between the 5′ end of the intron and the exon[11]. The most recent method to capture lariats is co-transcriptional lariat sequencing (CoLa-seq), a technique that utilizes the extension of primers attached to the 3′ end of lariat products to map branchpoints[12].

The lariats identified by all these techniques measure lariats that escape degradation. As many introns contain multiple branchsites[13] and the efficiency of debranching is sequence-dependent[14], the dominant branchpoint detected through sequencing is not necessarily the dominant branchpoint used by the spliceosome. Dbr1 exhibits a bias for adenosine branchpoints in biochemical debranching assays[14]. The removal of Dbr1 through genetic knockout would aid efforts to identify branchpoints by reducing the degradation rate and sequence bias. While DBR1 disruption experiments have been performed multiple times in yeast[5,15], previous attempts to create DBR1 null mutants in vertebrate models have been unsuccessful. The breeding of DBR1 -/- mice resulted in non-viable embryos[16], and a CRISPR library used to introduce a variety of mutations into DBR1 in cell culture found a very strong depletion of cells that had taken up nonsense and frameshift mutations relative to other types of mutations[17].

Here we report the creation of a DBR1 knockout human cell line and use it to demonstrate that DBR1 encodes the sole debranching activity in 293T cells. Reporter assays, single-cell FISH, and lariat sequencing are used to define the life cycle of the excised lariat in the cell. Analysis of lariats recovered from our DBR1 knockout cell line reveals sequence signatures of Dbr1 specificity at both the branchsite and 5′ splice site. Co-immunoprecipitation (co-IP) and mass spectrometry identified Dbr1 binding partners, many of which are core spliceosomal factors. The location of these interacting proteins in the spliceosome's structure suggests the site through which Dbr1 gains access to the branchpoint. Recruitment of Dbr1 was also demonstrated through analysis of AQR, an RNA binding protein that binds proximal to the branchpoint and whose depletion partially phenocopies the lariat accumulation seen in the DBR1 knockout. In addition to elevating lariats, loss of Dbr1 activity increases exon skipping. Timestamp labelling experiments of lariats suggest a delay in spliceosome recycling. The effect on snoRNAs was modest, and intronic miRNA expression increased. Our final conclusion is the increase in exon skipping in Dbr1 null cells is caused by a kinetic delay mechanism in splicing caused by the retention of late spliceosomal components on stabilized lariats.

## Results

### The predominantly nuclear Dbr1 supplies all debranching activity in 293T cell extract

Currently, annotated branchsites are inferred from lariat-seq which samples the steady state pool of cellular lariats. The identity of lariats in this pool is influenced both by the sequence requirements of branchsite selection in splicing and the sequence specificity of the debranching enzyme Dbr1. To decouple these two processes, a commercial homologous recombination CRISPR kit was used to knock out DBR1 in 293T cells. Multiple clones were isolated and screened for recombination events (Supplementary Fig. 1A). To test for potential off-target mutations, we performed whole genome sequencing on 293T and C22 cells. The resulting reads provided 46.7x coverage for 293T and 45.9x coverage for C22. After read filtering, mapping and variant calling, 100,495 SNP and indel variants were found to be present in C22 but not in 293T. Using the Cas-OFFinder software, we identified 1458 predicted off-target sites. However, none of these sites overlapped with any of the C22 variants, indicating the absence of off-target mutations within this clone. To test if a disruption was sufficient to deplete Dbr1, whole cell extract was prepared. Western blot analysis visualized moderate Dbr1 expression in C9 but greatly reduced (C19) or absent (C22) expression in the DBR1 null clones confirming the PCR screening (Fig. 1A). Presumably, the faint band reactive to Dbr1 antibodies (C19 lane) represents an inactive missense mutant that arose through non-homologous end-joining. We tested the extracts in a cell-free debranching assay that utilized a quenched fluorophore in a synthetic branched RNA oligonucleotide[18]. While C9 extract debranched at near wildtype levels, the debranching activity of C19 and C22 was indistinguishable from the no-extract (i.e. null) control (Fig. 1B). The Dbr1 status of this cell line was further characterized by immunohistochemistry. Dbr1 was localized to the nucleus in wild type 293T cells but undetectable in the DBR1-negative cell line (C22, Fig. 1C). To measure the sub-cellular localization of lariats, single molecule FISH was designed to visualize the location of the Taok2 intron 13 in wild-type and Dbr1-depleted (C22) cells (Fig. 1D). As expected, the overall number of lariats increased two-fold in the knockout clone but also shifted to the cytoplasm. Taken together, these results suggest a primarily nuclear Dbr1 encodes the sole debranching activity in 293T cells. Lariats that escape debranching in the nucleus accumulate in the cytoplasm and the null C22 mutant line has an elevated level of cytoplasmic lariats.

### The DBR1 knockout cell line C22 is 20-fold less effective in lariat turnover

As the principal role of Dbr1 is to debranch lariats, the initial search for Dbr1 depletion phenotypes focused on measuring lariat levels using two RNA-Seq based methods developed in the Fairbrother lab. Consistent with the cell-free debranching assay, lariats are ~20-fold enriched in both C19 and C22 as measured by lariat-seq (Fig. 2A). As branchpoint readthrough is inefficient and sequence-dependent, it cannot be assumed all branchpoints are detected with equal efficiency[19]. To gain further quantitative information on lariat enrichment, we applied the ShapeShifter algorithm to read coverage data from C19 and C22 and identified the subset of introns in each cell line which were classified as having coverage curves indicative of lariat accumulation[11]. For each intron set, we then calculated changes in total normalized intronic read coverage between DBR1 knockout and wildtype samples. Consistent with the observed lariat read level increases upon DBR1 knockout, this analysis revealed sizeable increases in normalized intronic coverage for these lariat-associated introns (Fig. 2B). While a small set of introns decreases in coverage relative to wildtype, this reduction is due to their high expression values in the wildtype rather than any particular reduction in coverage relative to other introns within the DBR1 knockout

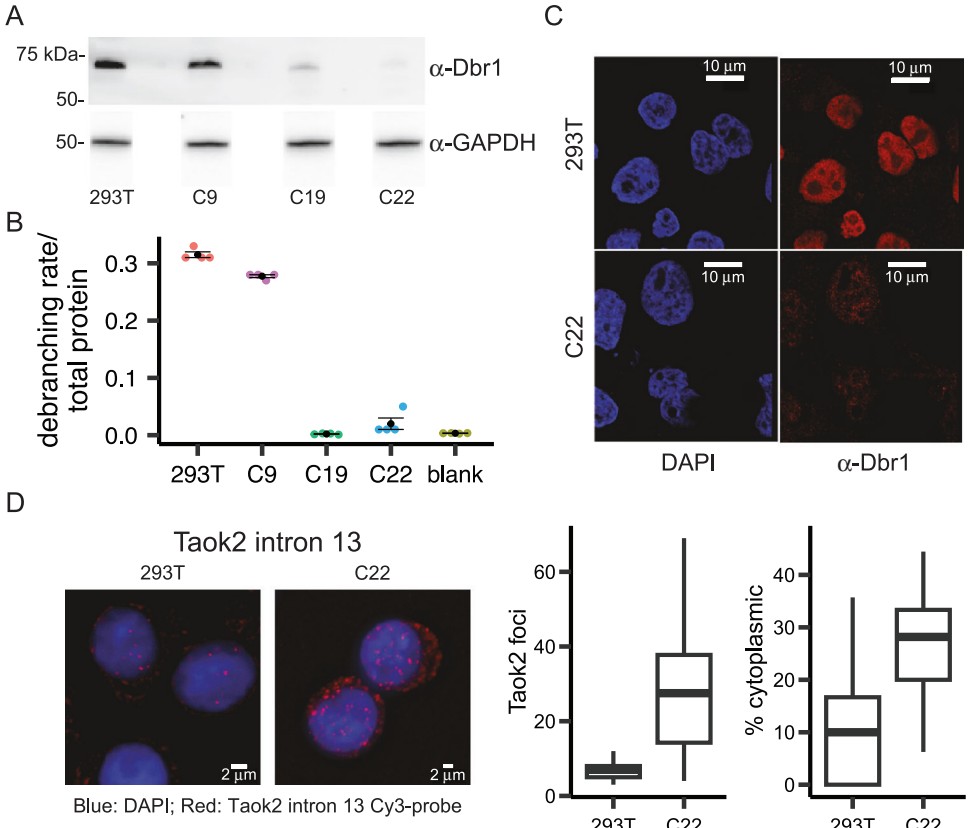

**Fig. 1 | The predominantly nuclear Dbr1 supplies all debranching activity in 293 T cell extract. A** Dbr1 protein levels in wild type 293T and *DBR1* knockout cells lines (C9, C19, C22) assayed by western blot. **B** Fluorogenic debranching activity assay of extracts from C9, C19, and C22 cells compared to 293T control lysates (*n* = 4 independent experiments for each cell line; sample values shown as colored points; mean ± SE shown in black). **C** Fluorescent immuno-microscopy of 293T and C22 cells, using DAPI stain and α-Dbr1 polyclonal antibodies. **D** FISH analysis of

Taok2 intron 13 in 293T and C22 cells (left), and quantification of cytoplasmic and nuclear intron foci (right; *n* = 35 cells for 293 T, 58 cells for C22; center line represents median; lower and upper bounds of the box represent the 25th and 75th percentile, respectively; lower and upper whiskers extend to the smallest or largest value no further than 1.5× the inter-quartile range from the 25th or 75th percentile value, respectively; outliers not shown). Source data are provided as a Source Data file.

(Supplementary Fig. 2). A large amount of variation in the degree of lariat stabilization is also observed. About 10% of the lariats recovered from *DBR1* knockout cells experienced a greater than 32-fold enrichment relative to the level observed in a wildtype background. Comparison of lariats extracted from *DBR1* positive and negative backgrounds reveals Dbr1 specificity. The sequence logos generated from the branchsites of recovered lariats shows the expected YTNAY branchpoint motif in C22 but not 293T cells (Fig. 2C). There is a strong polypyrimidine tract signal downstream of the branchpoint from *DBR1* knockout samples, peaking between positions +10 and +15 relative to the branchpoint. The sequence differences between 293T and C22 lariats also indicate Dbr1 specificity for parts of the 5' splice site. Relative to lariats from the wildtype, those from C22 cells are enriched for 'G' at positions 2 nucleotides (Chi-square, p = 1.74e-13) and 4 nucleotides (Chi-square p = 0.0016) downstream of the 5' splice site (Fig. 2D). Consistent with Dbr1 specificity for 5' splice site sequences, annotated U12 introns[20] (which do not contain the 5' motif enriched in C22 lariats) exhibit only ~60% of the lariat increase observed in U2 introns from the same genes upon *DBR1* knockout (Fig. 2E). Over 50,000 branchpoints in total were identified from RNA-Seq of *DBR1* knockout cells (Fig. 2F), and the lariats captured in a *DBR1* knockout background are more than two-fold enriched for 'A' branchpoints (Fig. 2G). This increase in the proportion of 'A' branchpoint lariats represents a molecular signature of lower Dbr1 activity consistent with earlier studies on the enzymatic specificity of Dbr1[14]. To characterize the relationship between branchsite motif and splicing activity, we performed a massively

parallel reporter assay that tested every hexamer's ability to function as a branchsite in the context of APRT intron 2 (Supplementary Fig. 3). Annotating the branchpoints from both *DBR1* knockout and wildtype contexts with the functional activity scores learned from our splicing assay indicates that branchpoints mapped in the Dbr1-deficient environment are more likely to be used than those found in the wildtype (Fig. 2H). The results of the BP functional assay taken together with the characterization of lariats recovered from a Dbr1-deficient context demonstrate how the branchpoint sequence plays a dual role in splicing catalysis and determining lariat stability.

## Dbr1 interacts with spliceosomal factors positioned around the branchpoint in the intron lariat spliceosome (ILS)

To better understand the mechanism of debranching, a semi-quantitative mass spectrometry approach was used to define the network of Dbr1 interacting partners. Briefly, FLAG-tagged Dbr1 was expressed in 293T cells (Fig. 3A, left). Coomassie staining showed Dbr1 complexes were efficiently isolated through FLAG antibody conjugated magnetic beads, and the components of these complexes were analyzed by mass spectrometry (Fig. 3A, right). 120 proteins were identified as highly enriched in the eluate above control (Supplementary Data 1). As the immunoprecipitation was performed without nuclease treatment, these interactions could be indirect and mediated by either nucleic acids or proteins. Analysis of these interactors using the Gene Ontology and Reactome databases revealed an enrichment for many RNA processing pathways including rRNA and ncRNA

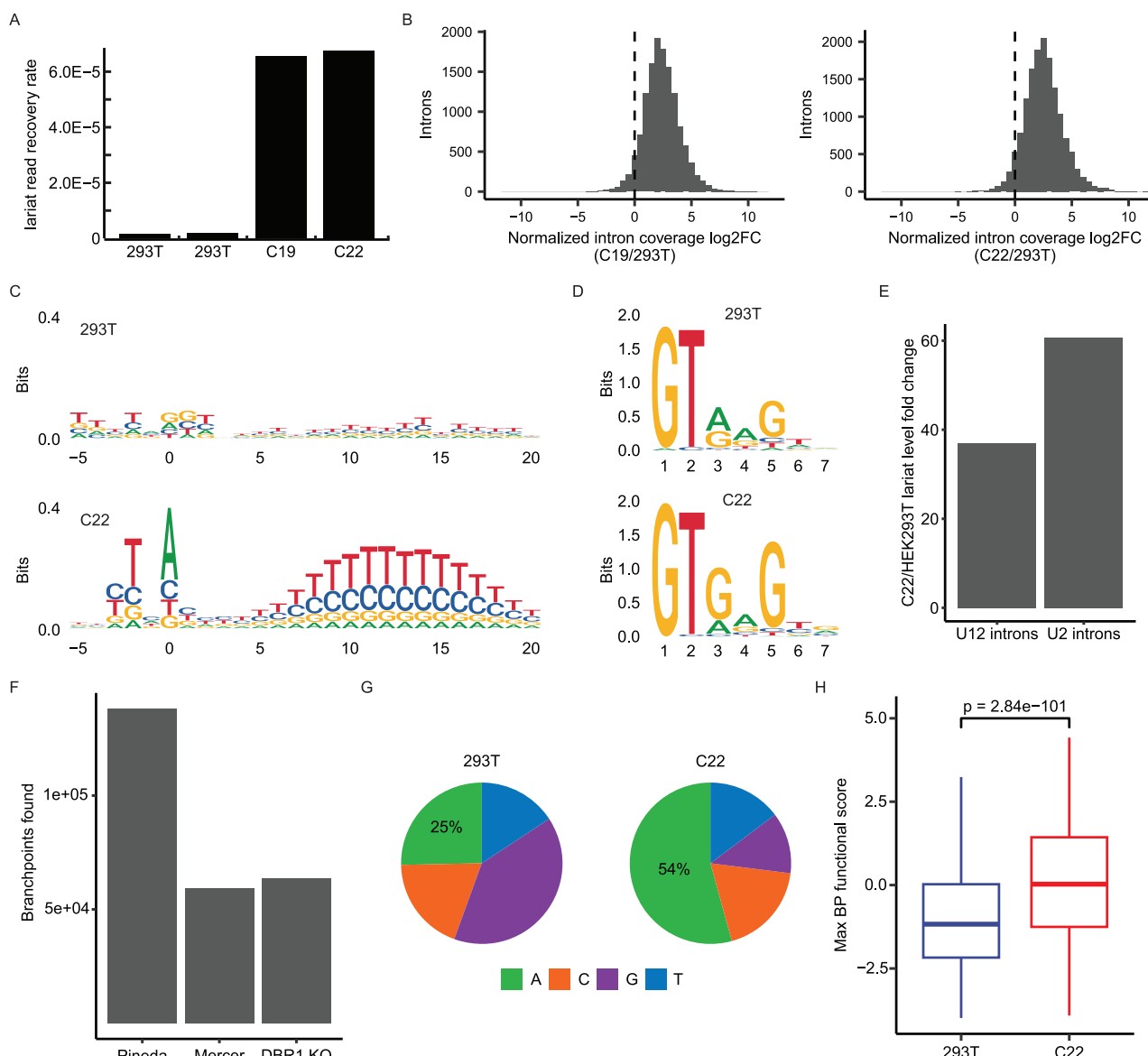

**Fig. 2 | The *DBR1* knockout cell line C22 is 20-fold less effective in lariat turn-over. A** The lariat read recovery rate (lariat reads/total mapped reads) in C19, C22 and two 293T control samples. **B** Fold change between *DBR1* KO and wild type samples in the coverage of individual introns that were classified by ShapeShifter[11] as exhibiting lariat accumulation. **C** Sequence logo of branchsites from lariat reads recovered in 293T and C22 samples from introns with a single recovered branchpoint. **D** Sequence logo of the top 100 5′ splice sites ranked by lariat read counts in 293T and C22 samples. **E** Change in lariat levels between C22 and 293T samples for annotated U12 introns as well as U2 introns from the genes containing U12 introns. **F** Tally of branchpoints reported in previous mapping studies (Pineda[13] and

Mercer[10]) and those found in *DBR1* KO cell lines. **G** Branchpoint nucleotide composition of lariats reads recovered in 293T and C22 samples. **H** Distribution of the maximum branchpoint functional score in an 11 bp window centered on branchpoints from lariat reads recovered in 293T ($n = 1374$ branchpoints) and C22 samples ($n = 15829$ branchpoints; $p$ value from two-sided t-test; center line represents median; lower and upper bounds of the box represent the 25th and 75th percentile, respectively; lower and upper whiskers extend to the smallest or largest value no further than 1.5x the inter-quartile range from the 25th or 75th percentile value, respectively; outliers not shown). Source data are provided as a Source Data file.

processing (Supplementary Data 2). Dbr1 interacts with members of the U5, U2, NTC, and NTR spliceosomal sub-complexes as well as Cwf19L, the human homolog of yeast Drn1 (Supplementary Data 1). Dbr1 has not been isolated with the spliceosome and is not part of existing structures. However, the location of the Dbr1-interacting components of the spliceosomal C complex (which contains the intron intermediate) and the post-catalytic intron lariat spliceosome (ILS) complex (which contains the excised intron) reveals insight into how and when Dbr1 approaches the branchpoint (Fig. 3B; C complex - PDB 6zym and 7a5p, ILS complex - PDB 6id1[21,22]). Dbr1 interacts with proteins proximal to the branch site in both the catalytic and post-catalytic complex. However, in the cryo-EM structures, the branchpoint is not

accessible from this Dbr1-interacting side of the structure until the spliceosome transitions to the post-splicing ILS complex. This result suggests Dbr1 acts on the fully excised intron and not lariat intermediates, consistent with Dbr1's more extensive association with second-step splicing factors (6 interactors - PLRG1, PRP8, XAB2, PRP19, RBM22, AQR) relative to first-step splicing factors (3 interactors - PRP8, PRP19, RBM22) (Supplementary Data 1) as well as crosslinking data from yeast localizing Prp8, Dbr1 and Drn1 to the branch site[23]. In addition to the core components of the spliceosome, there are interactions with other splicing factors that bind introns and could potentially recruit Dbr1 to lariats. Cross referencing the set of Dbr1-interacting proteins with publicly-available eCLIP binding data[24],

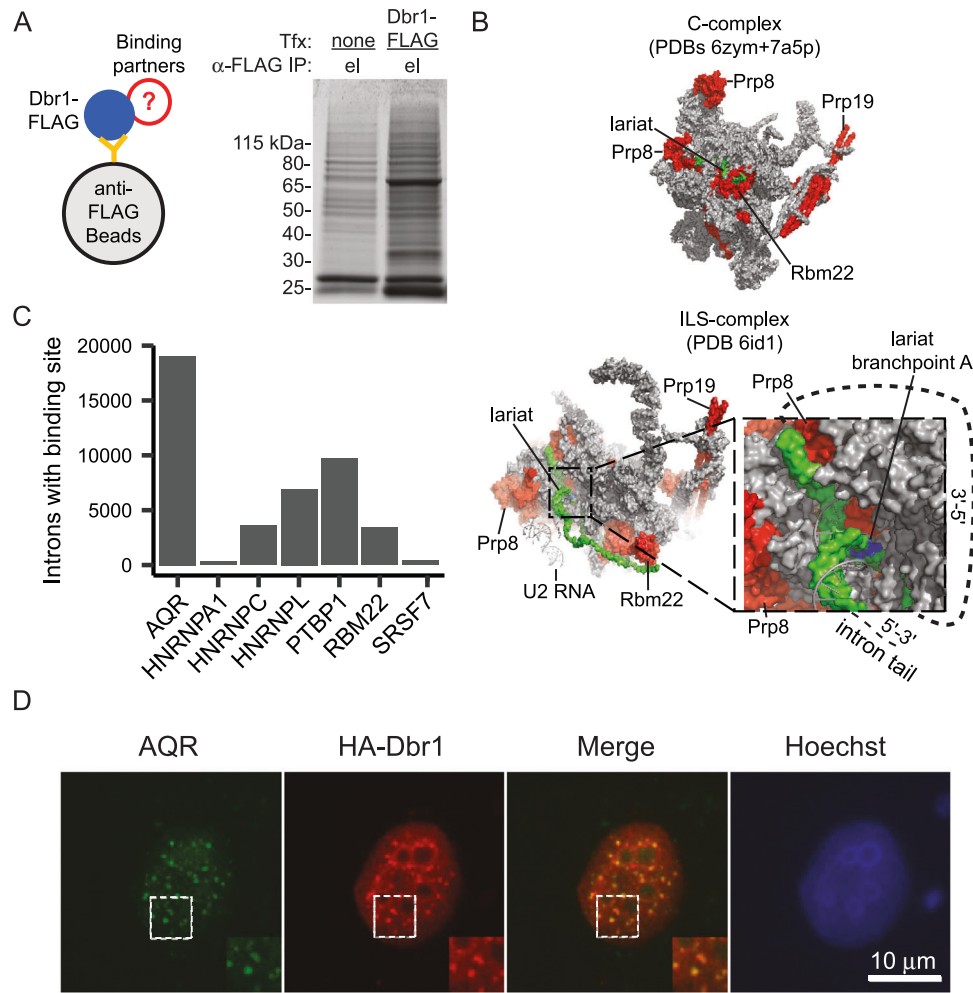

**Fig. 3 | Dbr1 interacts with spliceosomal proteins and other splicing factors.**
**A** Dbr1-FLAG and binding partners were isolated through co-immunoprecipitation with anti-FLAG magnetic beads. The eluate shows a clear band for Dbr1 on a Coomassie-stained gel (*n* = 3 independent replicates for both non-transfected control cells and Dbr1-FLAG transfected cells). **B** Significant interactions between Dbr1 and spliceosome factors Prp8, Prp19, and Cwc2 (red) were detected, and these factors are adjacent to the lariat (green) in cryo-EM structures of the C and intron lariat spliceosome (ILS) complexes. After transition to the ILS complex, the lariat becomes accessible to Dbr1. **C** The count of human introns with an eCLIP binding site from ENCODE for Dbr1 co-IP partners that are RNA-binding proteins. **D** Immunofluorescence using anti-AQR and anti-HA antibodies in U2OS cells shows Dbr1 and AQR co-localize to nuclear speckles (*n* = 3 independent replicates for each treatment). Source data are provided as a Source Data file.

reveals AQR as the most intronic binder (Fig. 3C). The AQR-Dbr1 interaction was identified by pulldown with an exogenous expressed tagged Dbr1. AQR and Dbr1 also co-localize in vivo in nuclear speckles (Fig. 3D).

## AQR recruits Dbr1 to branchpoints

To evaluate the ability of AQR and other intron-binding proteins to recruit Dbr1 to branchsites, introns were scored for their sensitivity to Dbr1 depletion. We reasoned that if Dbr1 was actively recruited to certain introns via an RBP, introns with binding sites for that RBP would show a stronger dependence on Dbr1 for their turnover (i.e. a larger increase in lariat reads in Dbr1-deficient cells compared to non-targeted introns). For each RBP identified in the Dbr1 pull down assay, the set of introns that contained at least one RBP binding site (i.e. eCLIP enrichment) was identified[24]. As a control, the same number of introns was sampled from the set of introns with no reported eCLIP sites for any of the analyzed RBPs. The normalized lariat levels in wildtype and *DBR1* knockout samples were then calculated for each intron set, and the lariat enrichment of the *DBR1* knockout relative to wildtype was compared between the bound and non-bound intron sets. Introns bound by AQR, RBM22 and SRSF7 exhibited significant increases in

lariat levels upon Dbr1 depletion relative to introns that were not bound by these RBPs (Fig. 4A). Overlapping the RBP binding positions with the 5'ss and branchpoint locations identified in the lariat data indicates that the intronic AQR eCLIP sites that were associated with the largest lariat response upon Dbr1 depletion are enriched within 50 nucleotides of the branchpoint along the circular topology of the lariat structure (Fig. 4B). No such relationship is observed for the RBM22 and SRSF7 sites. In addition to the co-localization of Dbr1 and AQR in nuclear speckles, the AQR-Dbr1 interaction predicted by mass spectrometry was further confirmed by western blot analysis of both Dbr1 and AQR immunoprecipitations (Fig. 4C, top). Furthermore, if AQR recruits Dbr1, the loss of AQR should also phenocopy the loss of Dbr1 (i.e. increased lariat reads and enrichment of 'A' branchpoints). Knockdown of *AQR* through siRNA lead to a reduction of AQR without affecting the level of Dbr1 (Fig. 4C, bottom). Despite wildtype Dbr1 levels, lariat reads increased in *AQR* knockdown samples relative to untreated controls, consistent with the hypothesis that AQR recruits Dbr1 to specific introns for debranching (Fig. 4D). Comparing AQR eCLIP sites to the lariats mapped from both the *DBR1* knockout and *AQR* knockdown shows that the presence of an AQR site in an intron increases the likelihood of lariat recovery in these samples

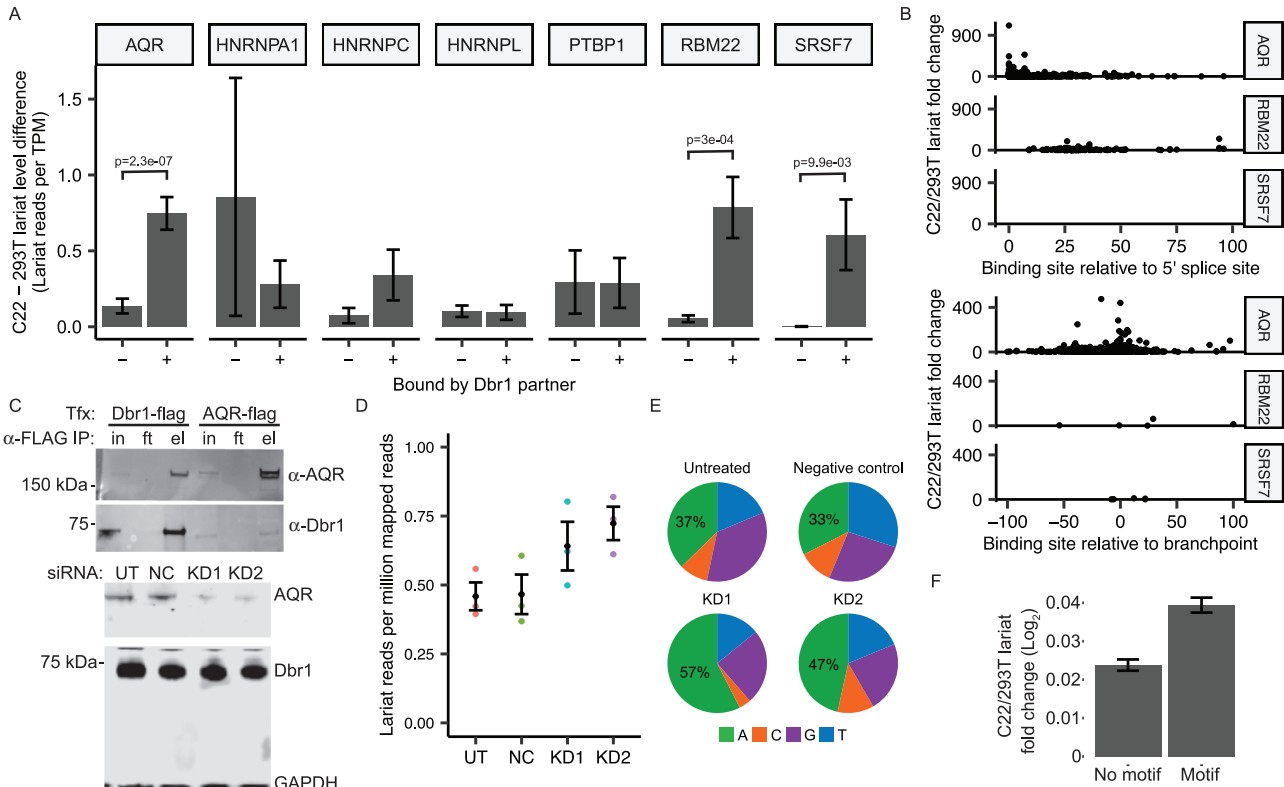

**Fig. 4 | AQR recruits Dbr1 to branchpoints. A** For each Dbr1-interacting RNA-binding protein, the change in normalized lariat levels between C22 and 293T samples for introns with and without reported eCLIP binding sites (mean ± SE shown; $n = 15087$ introns for AQR, 230 introns for HNRNPA1, 2462 introns for HNRNPC, 4322 introns for HNRNPL, 6330 introns for PTBP1, 2576 introns for RBM22 and 280 introns for SRSF7; $p$-values from two-sided t-test). **B** For the three RBPs with significant changes in (**A**), the change in lariat levels between C22 and 293T samples is compared to the location of RBP binding sites relative to the 5′ splice site and branchpoint. **C** Reciprocal co-IP of Dbr1 and AQR in 293T cells (top;

$n = 3$ independent replicates), and Dbr1 and AQR levels in untreated (UT), non-targeted control (NC) and two *AQR*-targeted siRNA knockdown samples (KD1 and KD2, bottom; $n = 3$ independent replicates). **D** Lariat mapping of RNA-seq data from control and *AQR* knockdown samples ($n = 3$ independent replicates shown in color with mean ± SE in black). **E** Branchpoint nucleotide composition of lariat reads recovered in control and *AQR* knockdown samples. **F** Fold-change in lariat levels between C22 and 293T samples for introns containing the AQR binding motif learned from eCLIP peak sequences (mean ± SE shown; $n = 17304$ introns for both motif and no motif categories). Source data are provided as a Source Data file.

(Supplementary Fig. 4). Additional confirmation that this elevation of lariats is Dbr1-dependent can be found in the relative increase in the Dbr1-preferred 'A' branchpoint substrate following AQR depletion (Fig. 4E). The GC-rich motif enriched in AQR binding regions was mapped to all annotated human introns and, like the eCLIP binding sites, was predictive of increased sensitivity to Dbr1 (Fig. 4F). The clustering of AQR sites in Dbr1-dependent introns 50 nucleotides downstream of the 5′ splice site or within 50 nucleotides of the branchpoint implies a model wherein recruitment can occur along the lariat loop at either side of the branchpoint.

**Dbr1 enhances splicing of cassette exons**

Our proteomic analysis revealed Dbr1 interacts with many splicing factors (Supplementary Data 1). To explore the role of Dbr1 in splicing, total RNA was extracted from wildtype and Dbr1-deficient cells. Isoform analyses were performed by rMATS to identify splicing events that differed between *DBR1* knockout and wildtype states. The loss of Dbr1 affected the inclusion of cassette exons most strongly. Dbr1 disruption caused exon skipping similar to the relatively non-specific splicing inhibitor Pladienolide B (Pla-B, Fig. 5A). An analysis of individual events depicts this influence as a moderate effect across many exons (Fig. 5B). Dbr1 is unlikely to be a splicing factor as a) it is only known to function post-splicing, b) it does not recognize linear RNA and c) it is not a component of the spliceosome. Dbr1-deficient cells have previously been found to have a higher proportion of spliceosomal components existing in post-splicing complexes relative to pre-

splicing complexes, indicating a potential defect in spliceosome recycling[25]. We reasoned that a delay in recycling of spliceosomes caused by a loss of Dbr1 could reduce the pool of free spliceosomes ready to engage nascently transcribed introns. As splicing is co-transcriptional, new splice sites become available as downstream exons are synthesized, and a delay in splicing could increase the possibility of splicing to a newly synthesized downstream 3′ splice site (i.e. exon skipping). Analysis of the ratio of spliced-to-unspliced reads shows an increase in unspliced product for introns in *DBR1* knockout cells relative to wildtype (Fig. 5C). There is a strong enrichment of A branchpoints in introns upstream of exons skipped in the *DBR1* knockout (Fig. 5D), suggesting introns susceptible to skipping are under selective pressure to rapidly debranch. To assess whether elevated lariats were specifically responsible for the shift toward pre-mRNA observed in *DBR1* knockout cells, we performed an RNA timestamp experiment by introducing chimeric fusions of ADAR paired to late spliceosomal proteins (PPIE or RBM22) into *DBR1* knockout cells[26]. A delay in spliceosomal recycling should prolong contact between the ADAR fusions and the lariats, increasing the degree of editing. Lariat PCR across a panel of 10 introns sensitive to Dbr1-induced exon skipping was used to assay the degree of editing in wild type and *DBR1* knockout cells. *DBR1* knockout samples had 2- to 3-fold higher editing rates relative to wildtype cells when transfected with the two ADAR-RBP fusion constructs (Fig. 5E). This result demonstrates that spliceosomal components remain associated with post-splicing lariats two to three times longer in the absence of debranching activity.

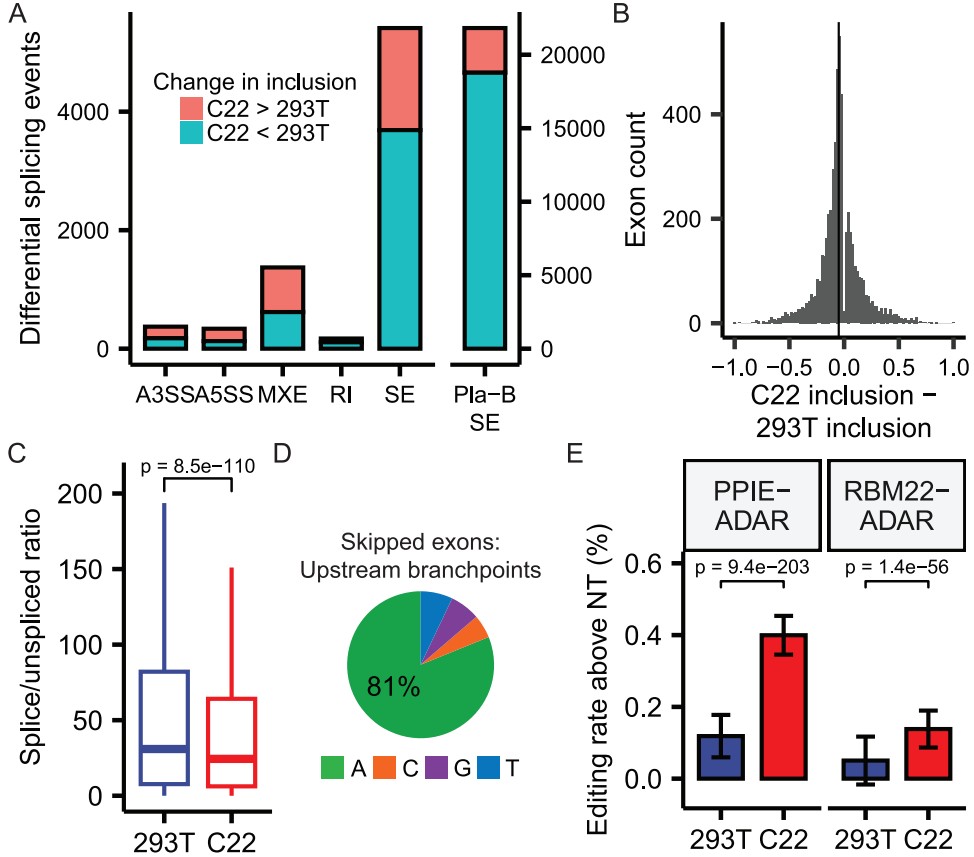

**Fig. 5 | Dbr1 enhances splicing of cassette exons. A** The count of differential splicing events computed by rMATS (FDR < 0.05) between C22 and 293T samples for alternative 3′ splice site (A3SS), alternative 5′ splice site (A5SS), mutually-exclusive exon (MXE), retained intron (RI), and skipped exon (SE) events. The skipped exon counts observed in a Pladienolide B splicing inhibition experiment are shown for comparison (right panel). **B** Change in inclusion between C22 and 293T samples for differentially-spliced skipped exons (FDR < 0.05). Vertical line indicates the median inclusion change (−0.05). **C** Distribution of the ratio of spliced (exon-exon) to unspliced (exon-intron and intron-exon) reads for introns in 293T and C22 samples (n = 59723 introns; p-value from two-sided t-test; center line represents median; lower and upper bounds of the box represent the 25th and 75th percentile, respectively; lower and upper whiskers extend to the smallest or largest value no further than 1.5x the inter-quartile range from the 25th or 75th percentile value, respectively; outliers not shown). **D** Branchpoint nucleotide composition in introns upstream of exons that are differentially skipped in C22 samples. **E** The percentage of edited adenosine bases in 10 sequenced lariats from cells transfected with PPIE-ADAR or RBM22-ADAR fusion proteins after correction with the base editing rate observed in non-transfected (NT) cells (mean ± SE shown; for 293T, n = 213 bases for non-transfected sample, 212 bases for PPIE-ADAR-transfected sample, 204 bases for RBM22-ADAR-transfected sample; for C22, n = 441 bases for non-transfected sample, 407 bases for PPIE-ADAR-transfected sample, 400 bases for RBM22-ADAR-transfected sample; p-value from Pearson's Chi-squared test). Source data are provided as a Source Data file.

## Dbr1 function is mostly restricted to intron splicing and turnover

Dbr1 depletion affects splicing and lariat levels. To search for additional *DBR1* null phenotypes, a broad range of genomic analysis were undertaken. Dbr1 does not appear to have a directional effect on transcripts levels (Fig. 6A/B). Lariats are known to act as expression vectors for small RNAs including microRNA and snoRNA[27–29]. An analysis of microRNA in Dbr1-deficient and wildtype cells revealed a 10% increase in the proportion of intronic miRNA among those with increased expression upon Dbr1 depletion (Fig. 6C, Supplementary Fig. 5A). Surprisingly, cloning small RNA from C22 and wildtype cells showed snoRNA levels were not strongly affected by the depletion of Dbr1, but an analysis of the location of expressed snoRNA genes showed a close spatial relationship between the boundary of the snoRNA gene and the lariat-seq determined branchpoint (Fig. 6D, Supplementary Fig. 5B). To test the potential for Dbr1 to act on other RNA linkages, we utilized several RNAs as competitors in the cell free debranching assay. cGAMP and a noncanonical (G-ppp-G) 5′ cap analog had minimal effects on the debranching rate, but a canonical (G-ppp-A) 5′ cap analog inhibited debranching in a concentration-dependent manner (Supplementary Fig. 6A). While this suggests the presence of

an interaction between Dbr1 and certain 5′ caps, assays performed to test Dbr1's ability to hydrolyze capped RNA failed to show any such activity (Supplementary Fig. 6B/C). We conclude the direct consequences of Dbr1 LOF is a failure to debranch lariats, and the resulting excess of lariats causes exon skipping through a defect in spliceosome recycling.

## Discussion

While prior failures to knockout *DBR1* suggested it was an essential gene[16,17], 293T cells deficient in Dbr1 are viable. Similar to depletion studies in *S. pombe*[30], these cells exhibit growth defects (Supplementary Fig. 1B). Performing debranching assays and quantification of lariat read recovery in this *DBR1* knockout cell line demonstrates that *DBR1* encodes the sole debranching activity in 293T cells. The Dbr1 homolog Drn1 was found to co-IP with Dbr1; however, this study suggests its presence is not sufficient to contribute to debranching presumably because it lacks the metal center required for catalytic activity[23,31].

Analysis of the branchsites recovered from lariats in a Dbr1-deficient context reveals a dominant branch site motif that is markedly similar to the canonical motif (Fig. 2C). This similarity underscores the

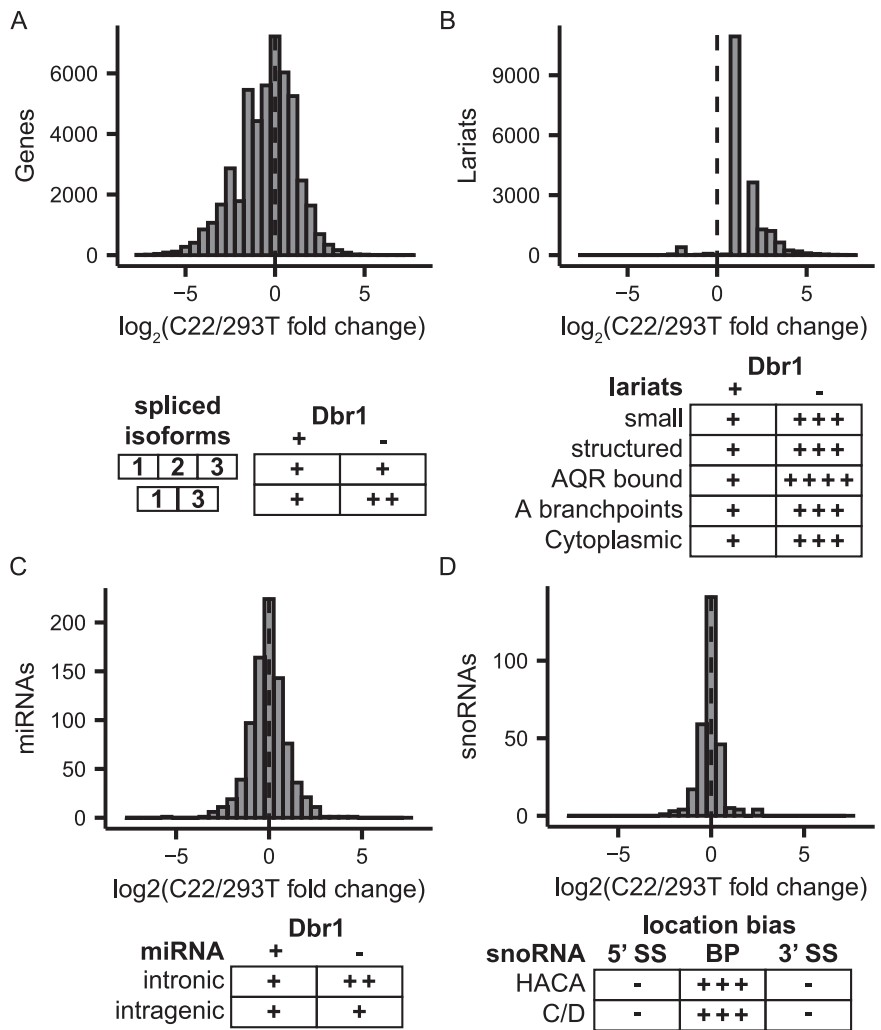

**Fig. 6 | Phenotypes related to Dbr1 loss. A–D** Quantitative (top) and qualitative (bottom) changes due to *DBR1* KO in transcripts (**A**), lariats (**B**), miRNAs (**C**) and snoRNAs (**D**). Source data are provided as a Source Data file.

value of mapping branchpoints using lariat-seq in a *DBR1*-null context as the sequence specificity of Dbr1 leads to preferential hydrolysis of branchsites that are commonly utilized by the spliceosome. In addition to branchsite sequence affecting Dbr1 activity, our results show that sequence features of the 5′ splice site are also likely influencing Dbr1 specificity. The most abundant 5′ splice sites from lariats recovered in *DBR1* knockout samples are significantly enriched for 'G' at two positions relative to wildtype lariats (Fig. 2D), and minor spliceosomal introns with alternate 5′ splice site motifs only undergo about half the lariat level increase upon *DBR1* knockout that major spliceosomal introns do (Fig. 2E). This discrepancy suggests that minor intron lariats are more reliant on Dbr1-independent pathways for degradation.

Dbr1 is unlikely to act alone as co-IP mass spectrometry reveals interactions with many other factors involved in RNA processing (Supplementary Data 2). These binding partners have placed Dbr1's molecular function within the structural context of the spliceosome. Dbr1 binds to proteins within the C complex spliceosome that contains the intron intermediate[22]. However, structures of this complex suggest the branchpoint of the intron intermediate is buried in the spliceosome until both steps of splicing are complete. It is not until the transition to the post-splicing ILS complex that the branchpoint becomes accessible to Dbr1, and a set of ILS factors that are spatially close to the branchpoint within this complex appear to interact with Dbr1[21].

Further analysis of Dbr1 binding partners in terms of their effect on lariat levels showed that introns with AQR binding sites had a strong lariat response upon *DBR1* knockout relative to matched introns without an AQR binding site. While similar increases were observed for introns bound by RBM22 and SRSF7, AQR was the only of these three RBPs that exhibited a relationship between the localization of its binding sites and an intron's response to *DBR1* knockout. The AQR eCLIP sites associated with the strongest response to Dbr1 depletion in terms of lariat levels tended to cluster within ~50 bp of the branchpoint (either downstream of the 5′ SS or upstream of the branchpoint), and knockdown of *AQR* mirrored the hallmarks of Dbr1 depletion (increased lariats and a shift towards A branchpoints in the recovered lariat reads). Taken together, these data suggest an interaction in which AQR interacts with Dbr1 in order to recruit it to particular lariats for debranching. Due to the absence of nuclease treatment in the immunoprecipitation protocol performed, this interaction could be mediated by nucleic acids. Furthermore, a recent publication demonstrates that Dbr1 is linked by the TTDN1 protein to the intron binding complex (IBC) of which AQR is a component[32]. Given this connection between Dbr1 and the IBC, the interaction between AQR and Dbr1 could also be occurring through any of the proteins within that complex. Recent work from the Yeo lab also localized AQR binding to lariats based on eCLIP RT extensions ending at the nucleotide position downstream of the branchpoint and the 5′ splice site[33]. In their survey of 150 RNA-binding proteins, AQR depletion was

found to cause the largest number of alternative splicing events of all the proteins surveyed, raising the question of how an RBP associated with a post-splicing complex could influence splicing.

Here, we seek to explain this observation with a kinetic delay model framed in the context of co-transcriptional splicing. Because splicing is co-transcriptional, a delay in splicing enables more downstream exons to be synthesized and become available for catalysis. The influence of kinetics on splicing was first proposed in the context of polymerase pausing[34]. Our analysis of splicing in *DBR1* knockout cells revealed a strong tendency towards exon skipping, and the set of exons that were differentially skipped had an enrichment for 'A' branchpoints in the upstream intron. This sequence feature may suggest selective pressure on these flanking introns to rapidly debranch in the wildtype state. Thus, Dbr1 may be influencing exon inclusion indirectly by degrading the lariat product and releasing late spliceosomal components back to the catalytic spliceosome. Base editor fusion experiments provide support for a defect in recycling as a higher level of editing is observed in lariats from *DBR1* knockout cells, indicating the targeting spliceosomal proteins PPIE and RBM22 were associated with the splicing product for longer. When recycling is delayed, a larger proportion of transcripts remain in the unspliced state while the transcriptional machinery synthesizes additional downstream competitor 3' splice sites. In some cases, these downstream exons may be able to compete for splicing thus resulting in the phenotype of exon skipping. Han et al. also found an excess of post-splicing complex but proposed that exon skipping occurs at exons with poorly-recognized splice sites[25]. In contrast, we found branchsites associated with exon skipping were good substrates for both splicing and debranching (81' 'A' branchpoints, Fig. 5D). Our model incorporates the co-transcriptional nature of splicing. We suggest a delay in splice site selection would simply leave the spliceosome with more choice as potentially favorable downstream 3' splice sites are continuously being transcribed and becoming available for exon skipping. This kinetic delay mechanism has been hypothesized to explain a similar problem in alternative splicing[35].

## Methods

### Critical resources
Details on the resources utilized in the course of this study including antibodies, peptides, commercial assays, cell lines, oligonucleotides and recombinant DNA are provided in Supplementary Table 1.

### Cell culture
Human cells (293T and U2OS obtained from ATCC) were cultured at 37 °C and 5% $CO_2$ in Dulbecco's modified eagle medium (Thermo Fisher Scientific) supplemented with 10% Fetal Bovine Serum (Gibco).

### CRISPR cell line construction
293T cells were cultured in DMEM (Thermo Fisher Scientific) and 10% Fetal Bovine Serum (Gibco) on 10 cm plates 24 h prior to transfection at 40% confluency. The following were transfected to each plate: 3.6ug *DBR1* sgRNA (5'- AGACGCTGGCGCTGGCAGAG-3') or scramble control (Origene), 3.6ug GFP-puro donor DNA (Origene), 15ug GeneArt Platinum Cas9 nuclease (Thermo Fisher Scientific) and 43ul Lipofectamine 2000 (Thermo Fisher Scientific). At 48 h post-transfection, cells were split 1:10, and again every 3 days (7 times in total). After which, cells were grown in 0.5ug/ml Puromycin for selection, and single colonies were expanded. Minute™ Total Protein Extraction kit (Invent Biotechnologies) was used, and *DBR1* knockout werewas confirmed on single colonies #19 (C19) and #22 (C22) by western blot with the following antibodies: rabbit polyclonal anti-Dbr1 (Proteintech, Cat no: 16019-1-AP) at 1:500 and mouse monoclonal anti-GAPDH (Santa Cruz Biotechnology, sc-47724) at 1:200. Cells from single colonies C19 and C22 were further validated with genomic PCR, and RNA was extracted using TRIzol™ (Thermo Fisher Scientific) per

manufacturer's instruction and sent to Genewiz for library preparation and sequencing.

### Genomic PCR
Genomic DNA was harvested from wild-type 293T, C9, C19, and C22 cell lines using PureLink™ Genomic DNA Miniprep Kit (Thermo Fisher Scientific K182001). PCR was performed using Phusion polymerase (Thermo Fisher Scientific F548L), and primers: Puro-forward (5'-CCTATGACCGAGTACAAGCCC-3') and right-homology arm-reverse (5'-GCGTACTATGGTTG TTTGACGTATG-3'), and GFP-reverse (5'-TAGGTGCCGAAGTGGTAGAAGC-3') and left-homology arm-forward (5'-CGTAATCATGGTCATAGCTGT TTCCTG-3'), and visualized on a 1% agarose/ethidium bromide gel.

### Whole genome sequencing and off-target analysis
293T and C22 were cultured in DMEM + 10% FBS. DNA was extracted and analyzed with 2x100bp whole genome sequencing. The resulting reads were trimmed of adapter content and low quality bases using fastp. Trimmed reads were aligned to the hg38 genome with bwa. Variants were called from read alignments using bcftools mpileup and bcftools call. Variants were filtered to remove low quality calls (quality score <10 or read depth <10). To assess potential off-target CRISPR editing, C22 and 293T variants were overlapped, removing any C22 variants that are also observed in 293T. A variant allele fraction of 0.05 was required, and variants overlapping with repetitive regions annotated by RepeatMasker were discarded. After these filters, 100,495 final C22 variants remained. The locations of these variants were compared to those of 1458 potential off-target sites predicted by Cas-OFFinder (allowing for up to 3 mismatches and 1 DNA or RNA bulge).

### Debranching assay and Western blot
293T, C9, C19, and C22 cell lines were cultured in DMEM + 10% FBS. Cell lysates were prepared with Roche Complete M lysis kit, EDTA free according to the manufacturer's protocol (Roche, 04719964001). Protein concentrations were measured with the BioRad protein assay kit and adjusted to 1 mg/ml. For the fluorescent debranching assay, the reaction conditions were 0.2 uM bRNA substrate AK88, 1/10 volume 1 mg/ml lysate, 50 mM HEPES pH 7, 1 mM TCEP, and 100 mM NaCl. Reactions were monitored with fluorescence intensity using a BMG PHERAstar plate reader, (488 nm excitation, and 520 nm detection). The linear portions of the progress curves were fit with a slope and plotted as RFU/sec/mg of total protein. Western blots of the lysates were performed with an anti-Dbr1 polyclonal antibody using standard protocols (Proteintech 16019-1-AP).

### Immunofluorescence microscopy
293T and C22 cells were fixed with 4% paraformaldehyde, permeabilized with 0.1% Triton X-100, blocked with 1% BSA in PBS, incubated with 1:25 dilution of anti-Dbr1 antibody (Proteintech 16019-1-AP) for 3.5 h at room temperature. Cells were washed 3x with PBS, incubated with 4 ug/ml of secondary goat anti-mouse Alexa Fluor 647 (Thermo Fisher Scientific, A-21246) for 45 min at room temperature, washed 3x in PBS, incubated for 5 min in NucBlue™ (Thermo Fisher Scientific R37606), and mounted on glass slides with Diamond Antifade Mountant (Thermo Fisher Scientific P36965) and imaged with a Nikon inverted fluorescent microscope.

### Fluorescence in-situ hybridization
To visualize lariat RNA species from intron 13 of the TAOK2 gene, Quasar570®-labeled smRNA-FISH probes were ordered from Stellaris RNA-FISH (LGC Biosearch technologies). smRNA-FISH was performed according to the manufacturer's instructions. Briefly, $1 \times 10^6$ cells were fixed in 3.7% (vol./vol.) formaldehyde and then permeabilized in 70% ethanol at 4 °C. For in situ hybridization, permeabilized cells were incubated with 1.25 μM probes in the hybridization buffer (10%

formamide, 10% dextran sulfate, 1 mg/mL E. coli RNA, and 0.2 mg/mL BSA in 2X SSC) at 37°C overnight. Afterward, cells were washed sequentially with wash buffer A (10% formamide in 2X SSC), wash buffer B (0.1% triton-x-100 in 2X SSC), wash buffer C (1X SSC with 1 μg/mL DAPI), and 1X PBS. Finally, these cells were plated onto the coverslip and mounted in ProLong™ Gold Antifade Mountant (Thermo Fisher Scientific). The mounted cells were imaged using Olympus IX81 inverted widefield microscope equipped with Hamamatsu Orca Flash 4.0 camera with 4 megapixels and a 100 × 1.45NA oil objective lens. Single RNA molecule counting was done using ImageJ, and statistical analysis was conducted in R.

### Identifying lariats by iterative read splitting

Total RNA sequencing of 293T, C19, and C22 was performed. Sequencing reads were processed using previously described lariat mapping scripts[8]. Briefly, reads are mapped to the human genome, and any reads that have a full linear mapping are discarded. Next, the unmapped reads are split into all possible head and tail segments of >15 bp, and these segments are mapped to the genome. Lariat reads are identified as those in which a tail segment maps to the starting (5′) nucleotides of an intron and the corresponding tail segment maps to the downstream portion of the intron. After finding these inverted read alignments, alignments where the downstream segment maps to the end of an intron (intron circles) are filtered out. The end of the downstream segment of each remaining read is taken to be the branchpoint location for the lariat from which the read originated.

### Identifying lariats by splice site mapping

An additional lariat mapping pipeline was implemented based on the method described in Pineda and Bradley, 2018[13]. First, reads are filtered out if they contain >5% ambiguous characters. Then, reads are mapped to the genome, and aligned reads are discarded. A mapping index is then created based on the unaligned reads, and a Fasta file containing the first 20nt of each annotated intron in the transcriptome is mapped to the unaligned reads. Reads are then identified where only one 5′ splice site maps to them and the alignment has no mismatches or indels. These reads are then trimmed of the sequence from the start of the 5′ SS alignment to the end of the read, and reads were the trimmed portion is <20nt are filtered out. The remaining trimmed reads are mapped to an index built from the last 250nt of every annotated intron. The trimmed read alignments are then filtered to only consider those with <=5 mismatches, <=10% mismatch rate, and no more than one indel of <=3nt. Then, for each trimmed read the highest scoring alignment was chosen after restricting to alignments in the same gene as the 5′ SS alignment and those with the expected inverted mapping order of the 5′ and 3′ segments. The end of this highest scoring alignment is then taken to be the branchpoint of the lariat the read is derived from.

### Mapping lariat reads in *DBR1* knockout samples

Sequencing data from samples of *DBR1* knockout as well as wild-type 293T cells were processed using lariat mapping pipelines described above. Lariat read counts per sample were aggregated by intron, and the fold change in lariat recovery between *DBR1* KO and WT was determined.

### Massively parallel splicing reporter assay of randomized branchpoint species

Using the sequence of APRT intron 2 and exon 3, a library of oligonucleotides was designed and inserted into a single-intron construct version of our previously published splicing assay[36]. Each species in the library has the following components: an 11 bp intronic barcode, the last 98 bp of APRT intron 2, the first 30 bp of APRT exon 3, and an 11 bp exonic barcode. The 6 bp window surrounding the branchpoint of

APRT intron 2 (from −3 to +2 relative to the branchpoint) was iteratively replaced by all 4096 hexamers. Each branchpoint hexamer was associated with four different pairs of intronic and exonic barcodes. Barcodes were prefiltered so that they: 1) did not contain a TAA, TGA, or TCA motif, which could act as an alternative branchpoint site, and 2) they were in the bottom 50% of barcodes when ranked by the sum of squared overlapping hexamer scores for five different hexamer-based scores of ESE or ISE activity from previously published minigene experiments: exon inclusion scores[37], and alternative 3′ss exonic, 5′ss exonic, 3′ss intronic, and 5′ss intronic scores[38]. This library was inserted into a reporter construct where the upstream sequence is the Cytomegalovirus (CMV) promoter and Adenovirus (pHMS81) exon with part of its downstream intron, and the downstream sequence is the bGH polyadenylation (polyA) signal. After assembling the constructs, samples of three replicates of the input library were taken for sequencing. The reporters were then transfected into 293T cells (ATCC) in three cell culture replicates using Lipofectamine 3000 (Invitrogen). RNA was extracted 24 h post-transfection using TRIzol™ (Thermo Fisher Scientific) and then DNase treated. Random 9-mers were used to generate cDNA with SuperScript IV Reverse Transcriptase (Invitrogen) followed by PCR (GoTaq, Promega). Following cDNA preparation, 2x150bp paired-end sequencing was performed using Illumina NovaSeq. The three input and output libraries were aligned using STAR to custom reference genomes containing either the input minigene sequences or the expected output spliced sequences (i.e. minigene sequence with expected intron removed) using the following STAR parameters: unspliced (--alignIntronMax 1), end to end (--alignEndsType EndToEnd), allowing up to 5 mismatches (--outFilterMismatchNmax 5), and only reporting uniquely aligned reads (--outFilterMultimapNmax 1). SAMtools idxstats was used to count reads for each input and output species. Splicing efficiency was calculated for each species $j$ after adding a pseudocount of 1 to all read counts as $(((o_j+1)/(i_j+1))/(\sum_k (o_k+1)/\sum_k (i_k+1)))$, where $o_j$ and $i_j$ are the output and input read counts for species $j$, and $\sum_k o_k$ and $\sum_k i_k$ are the sum of output and input read counts across all species. The mean and standard deviation of splicing efficiencies across four barcodes pairs in three replicates were calculated for each of the 4,096 branchpoint hexamers. The branchpoint functional score of a hexamer is defined as $log_2$(*mean splicing efficiency of hexamer's 4 barcode pairs*).

### Dbr1-FLAG immunoprecipitation

10-cm dishes of 293T cells were transfected with a Dbr1-FLAG plasmid[39]. 2 days post-transfection, cells were harvested with Roche Complete M lysis kit, EDTA free according to the manufacturer' protocol (Roche, 04719964001). Three dishes of Dbr1-FLAG transfected cells, and 3 dishes on un-transfected 293T cells were harvested. For each dish, 1 ml of 1 mg/ml lysate was used for immunoprecipitation with anti-FLAG M2 magnetic beads (Sigma, M8823) following the manufacturer's protocol. Samples were eluted with 50 ul of 2x SDS-PAGE loading dye and submitted for mass-spectrometry.

### Semi-quantitative co-immunoprecipitation mass spectrometry and analysis

Samples were separated ~1 cm on SDS-PAGE gels (Fig. 3A), and the entire lane divided into 5 gel slices. Peptides were extracted though in-gel trypsin digestion and analyzed on a mass spectrometer. Data were processed with Scaffold, and spectral counts exported to Excel. Filtering for hits was done in Excel by subtracting the average spectral counts of the 3 controls (293T + anti-FLAG beads) from the average of the 3 experiments (Dbr1-FLAG transfected 293T cells + anti-FLAG beads). Positive hits were those where the average of controls was <3, and the average of IP samples were >10. For proteins with 0 spectral counts, a pseudocount of 0.1 was used to allow for fold-change calculation. P-values were calculate using a two-sided t-test. The 120

proteins which passed filtering are presented in Supplementary Data 1. The list of 120 identifiers was analyzed with the following pathway analysis tools: Reactome, gSEA, and Gene Ontology, and Uniprot ID mapper.

## RNA-Seq sample preparation and siRNA knockdown

RNA from 293T and C22 cells was extracted with TRIzol™ (Ambion), total RNA libraries prepared using random 9mers for cDNA synthesis, followed by rRNA depletion, barcoding, and sequencing (Illumina). For siRNA knockdown of AQR, 293T cells were transfected with AQR-targeting siRNA s18725 and s18726 and control non-targeting siRNA 4390843 (Thermo Fisher Scientific) using Lipofectamine RNAiMAX Transfection Reagent (Life Technologies). Cells were harvested 48-hours post-transfection with TRIzol™.

## In vitro decapping immunoprecipitation

Commercially-available capped eGFP mRNA (Trilink, CleanCap EGFP mRNA L-7601) at 0.2 uM was exposed to 1 uM Dbr1, 100 U mRNA decapping enzyme (DCE, NEB M0608S), or 200 U yeast scavenger decapping enzyme (yDcpS, NEB M0463S), and incubated at 37 °C for 2 h. A 10 uL aliquot of the digestion was diluted into 1 ml of TBS supplemented with 1 U/uL murine RNAse inhibitor (NEB M0314), and 150 ug/ml GlycoBlue glycogen (Thermo Fisher Scientific, AM9516), and then 0.5 ml was incubated with 50 uL of anti-2,2,7 tri-methylguanosine agarose (Millipore NA02A) for 2 h at 4 °C with rotation. The immunodepleted flow-through sample was removed, and the beads were washed 3 times by centrifugation. Bound mRNA was eluted by incubation with 1 mM of 7-methylguanosine cap analog (Thermo Fisher Scientific AM8048). Input, flow-through, and elution samples were precipitated with ethanol, and resuspended in 50 uL of 2X RNA loading dye (NEB B0363), separated on a 6% urea-TBE gel (Thermo Fisher Scientific EC6265), and stained with Syber Gold (Thermo Fisher Scientific S11494) and visualized with a UV-transilluminator.

## In vitro XRN1 sensitivity assay

Firefly luciferase (fLuc) was transcribed with the T7 Express kit, capped with the Vaccinia capping kit, purified with a spin column (NEB T2030) and digested with 1 uM Dbr1, 1 U mRNA decapping enzyme (NEB M0608), or 1 U RPPH, with and without 1 U XRN1 (NEB M0338). After a 1 h incubation, samples were purified again with a spin column, and analyzed on a 1% agarose/TBE gel stained with ethidium bromide.

## In vitro Dbr1 inhibition assay

20 nM Dbr1 was combined with 0.2 uM AK88 and varying concentrations of G-G cap analog (NEB S1407), G-A cap analog (NEB S1406), and cGAMP (InvivoGen nacga23). Product development was measured as described in the "Debranching assay and Western blot" methods section. Resulting curves were fit with Graphpad Prism to obtain IC50 values using the model Y = Bottom + (Top-Bottom)/(1 + (X/IC50)).

## Quantifying lariat levels in introns bound by Dbr1 interactors

BED files containing the coordinates of RNA-binding sites identified by eCLIP experiments were obtained from ENCODE. The subset of proteins that co-IP with Dbr1 and have a known involvement in splicing was determined to contain AQR, HNRNPA1, HNRNPC, HNRNPL, PTBP1, RBM22, and SRSF7. For each of these proteins, the total number of intronic binding sites was obtained by intersecting the protein's binding sites with the introns annotated in the UCSC knownGene dataset (https://genome.ucsc.edu/cgi-bin/hgTables). For the set of introns bound by each RBP, the lariat read recovery rate (lariat reads per million mapped reads) was calculated for both the *DBR1* KO and WT conditions. For comparison, a control set of introns was constructed containing introns that lack these RBP binding sites but are in

the same genes as the RBP-bound set, and the lariat read recovery rate was also calculated for this set.

## Quantifying lariat levels of an AQR knockdown

Using the procedure described under "Identifying lariats by splice site mapping," lariat reads originating from introns annotated in the UCSC knownGene dataset were identified in samples from WT, AQR shRNA1-treated, AQR shRNA2-treated, and non-targeted shRNA-treated cells. The overall lariat level for each replicate was calculated as the number of lariat reads recovered per million reads mapping linearly to hg19.

## Lariat levels in introns containing an AQR motif

Using the AQR eCLIP sites and the procedure described in the section "Motif comparisons between RBNS and eCLIP" in Van Nostrand et al.[24], 5mers enriched within AQR binding sites were calculated and aligned to each other to create an AQR motif sequence logo. A position frequency matrix was derived from this motif and input to the PWMScan web server using default options to identify instances of the motif in the hg19 genome assembly[40]. Similar to the eCLIP analysis above, two sets of introns were then created, one with introns containing an AQR binding motif and another with introns in the same genes that have no motif. The lariat levels in the *DBR1* KO and WT conditions were then calculated for these two intron sets.

## Differential splicing analysis of *DBR1* knockout vs wild type

*DBR1* KO samples were compared to WT samples using rMATS with default settings. Significant differential splicing events were identified at an FDR of 0.05. For alternative cassette exons, the relationship between splicing outcome and observed branchpoint was assessed for exons which decreased in inclusion in the *DBR1* KO. Branchpoints recovered from *DBR1* KO lariat reads were intersected with the introns upstream of these skipped exons, and the branchpoint nucleotide distribution of this intron set was calculated.

## RNA timestamp with ADAR constructs

HyperADAR control was a gift from Michael Kharas (Addgene plasmid #166969; http://n2t.net/addgene:166969; RRID:Addgene_166969), MSI2-DCD ((MSI2-DCD was a gift from Michael Kharas (Addgene plasmid #166968; http://n2t.net/addgene:166968; RRID:Addgene_166968) and MSI2-ADA (MSI2-ADA was a gift from Michael Kharas (Addgene plasmid # 166967; http://n2t.net/addgene:166967; RRID:Addgene_166967) were purchased from Addgene. The MSI2 protein in the MSI2-ADA vector was replaced by GenScript with the protein coding sequence for the RBM22 protein to create ADAR-RBM22 and the PPIE protein coding sequence also replaced MSI2 to create ADAR-PPIE.

1 × 106 HEK293 cells were seeded into a 6-well dish and each well was transfected with 1 ug of ADAR-RBP (ADAR-RBM22, ADAR-PPIE), hyperADAR, or MSI2-DCD plasmid using Lipofectamine 3000 (Invitrogen) transfection reagent. 24 h post transfection RNA was extracted from cells using Qiagen RNeasy Spin Column Kit (Qiagen). RNA was converted into cDNA using SuperScript Reverse Transcriptase IV (Invitrogen) and random primer 9 (New England Biolabs). Lariat PCR was used to amplify cDNA using a nested amplification strategy for ten different targets. Validation of products was done using the QIAxcel ScreenGel Software. Products from each reaction were aggregated and PCR purified (Qiagen QIAquick PCR purification kit) and sent for Massachusetts General Hospital (MGH) Crispr Amplicon Sequencing.

## Small RNA sequencing

Total RNA was extracted by TRIzol™ from 293T WT and two *DBR1* KO 293T cell lines (Thermo Fisher Scientific). RNAs under 130 nt were enriched by BluePippin size selection gel (Sage Science) and processed using Illumina TruSeq Small RNA Library Prep Kit for sequencing.

## Analysis of Small RNAs in *DBR1* knockout cells

To identify miRNA-containing reads, exact matches to a database of miRNA obtained from QIAGEN CLC Genomics Workbench with the default setting were extracted from the sequencing reads of both the *DBR1* KO and WT 293T samples. The miRNA read counts were then normalized and tested using edgeR to identify miRNA which were significantly differentially expressed between *DBR1* KO and WT (FDR < 0.05). Each miRNA was classified as intronic based on whether their sequence exists within the introns defined by the CCDS annotation (intron sequences obtained from the UCSC hg19 table browser), and the proportion of miRNA that are intronic was assessed for the following sets of miRNAs: all miRNAs with sufficient reads for testing; miRNAs that were downregulated in C19, C22 or both *DBR1* KOs; miRNAs that were upregulated in C19, C22 or both *DBR1* KOs. To find snoRNA reads, the *DBR1* KO and WT sequencing data was aligned to a snoRNA database obtained from QIAGEN CLC Genomics Workbench with the default setting. snoRNA reads were then normalized and tested for differential expression using edgeR.

## Reporting summary

Further information on research design is available in the Nature Portfolio Reporting Summary linked to this article.

## Data availability

Source data are provided with this paper. The RNA-seq data generated in this study have been deposited in the GEO database. *DBR1* knockout RNA-seq data is available under accession codes GSE195586 and GSE195469. *AQR* knockdown RNA-seq data is available under accession code GSE195668. The mass spectrometry data generated in this study have been deposited in the MassIVE database under accession code MSV000092263. The hg19 genome was used for mapping. eCLIP binding data was obtained from ENCODE. Binding peak files can be accessed through the ENCODE Project website [https://www.encodeproject.org/] using the accession codes provided in Supplementary Data 3. Source data are provided with this paper.

## Code availability

All original code has been deposited to Github. The split read lariat mapping scripts can be accessed at https://github.com/jlbuerer/LaMIRA. The splice site lariat mapping scripts can be accessed at https://github.com/jlbuerer/lariat_mapping.

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

## Acknowledgements

National Institute of Health grants R01GM105681 and R01GM127472 supported L.B, N.E.C, A.W, C.D, A.J.T, J.W, R.S, S.R, C.L.L. and W.G.F. NIH grant F31 CA254143 supported B.A.T. N.M. was supported by NIH grants R01CA193318, R01CA227001, and P01CA092584, an American Cancer Society research scholar grant (RSG-18-156-01-DMC), the Centene Corporation, the Barnard Foundation, and the Alvin J. Siteman Cancer Center Investment Program, which is supported by the Foundation for Barnes-Jewish Hospital Cancer Frontier Fund and the National Cancer Institute, Cancer Support Grant P30 CA091842. We thank Dr. Susan T. Weintraub, University of Texas Health Science Center at San Antonio Mass Spectrometry Institutional Core Laboratory, for analysis of the co-IP samples, for data processing and for assistance with interpretation of the mass spectrometry results. S.T.W. was supported in part by NIH grant 1S10RR025111-01 for purchase of the Orbitrap mass spectrometer and the University of Texas System Proteomics Core Network for purchase of the Lumos mass spectrometer.

## Author contributions

L.B. conducted the splicing analysis, AQR binding and knockdown analysis, and contributed to writing the manuscript. N.E.C. validated the *DBR1* knockout, conducted and analyzed the co-IP, performed Dbr1 specificity assays, and contributed to writing the manuscript. A.W. performed the ADAR lariat timestamp assay. C.D. conducted the branchpoint splicing assay. B.A.T. and N.M. conducted the AQR localization assay. L.B. and A.J.T. conducted the lariat analysis of the *DBR1* knockout. J.W. conducted the *AQR* knockdown. R.S. constructed the *DBR1* knockout cell lines. S.R. designed the branchpoint splicing assay and analyzed its results. C.L. conducted the small RNA analysis and the knockout growth assay. Y.Z. and J.P.S. performed the TAOK2 localization assay. A.K. and M.J.D. provided the debranching reporter. N.M. contributed to writing the manuscript. W.G.F. contributed to experimental design and writing the manuscript.

## Competing interests

We disclose W.G.F. as the founder of Walah Scientific and member of the scientific advisory board for Remix Therapeutics. All other authors declare no competing interests.
