## [Peer Review File · Nature Communications]

The debranching enzyme Dbr1 regulates lariat turnover and intron splicingREVIEWER COMMENTS

Reviewer #1 (Remarks to the Author):

This manuscript reports a study of DBR1 in HEK 293T cells. The authors made the first viable DBR1 KO human cell line. Using these cells, they report that Dbr1 primarily functions as a debranching enzyme and affects intron splicing. Their findings are supported by sequence signatures of Dbr1 specificity at the branchsite and 5' splice site. The authors show that Dbr1 interacts with many core spliceosomal factors, and they focused on AQR. They observed that AQR binds proximal to the branchpoint and possibly recruit Dbr1 to its targets. It was also observed that loss of Dbr1 increases exon skipping. The authors designed a timestamp experiment to show that loss of Dbr1 may cause a delay in spliceosome recycling, thus explaining the exon skipping phenotype. Overall, this study generated very novel insights regarding the function and targets of Dbr1 and its role in splicing regulation. I only have a few points to help improve the manuscript.

1. The authors created the DBR1 KO cells via CRISPR, and generated two (nearly) KO clones C19 and C22. Have the authors confirmed absence of off-target effects? In most experiments, results using the C22 clone were shown, but not C19. Do the C19 results closely replicate those of C22?
2. Although the authors showed that DBR1 KO did not cause global gene expression shifts, there are a considerable number of genes whose expression levels are altered (Fig. 6A). Thus, in several analyses where intron coverage in sequencing was examined and compared between samples (e.g. Fig. 2B, 4C), a normalization against gene expression is needed.
3. In Fig. 2C, the branchpoint motif is relatively weak in 293T cells (upper panel). Could this indicate technical noise in lariat capture experiments? Or does it reflect a depletion of Dbr1 targets due to their rapid degradation?
4. What are the interpretations of the sequence preferences in 5' splice sites in Fig. 2D?
5. The model that AQR recruits DBR1 is interesting. More supporting evidence will be preferred, for example, the authors could mutate the AQR binding motifs and test for DBR1 recruitment.
6. Fig. 5C, although a low p value was shown, the effect size is very small. The authors should rule out the possible contribution of intron retention in this analysis.
7. Fig. 5E, the RBM22 results appear to show a smaller change in editing rates (C22 vs 293T) compared to PPIE. What are the authors interpretations? Does it reflect a limitation of the assay that ADAR activity may differ in different fusion contexts?

Reviewer #2 (Remarks to the Author):

General Overview:

In this study, Buerer et al. investigated intron removal during splicing, involving the formation and debranching of branched lariat RNAs. They generated a DBR1 knockout cell line and found that Dbr1 is the primary enzyme responsible for debranching in human cells. Dbr1 exhibited a preference for substrates with specific U2 binding motifs, suggesting sequencing-identified branch sites may not align with spliceosome preferences. Specific 5' splice site sequences interacting with Dbr1 were identified. The study proposed a model involving the intron-binding protein AQR in Dbr1 recruitment. Dbr1 depletion increased lariats and exon skipping, and spliceosome recycling was defective, prolonging lariat association and increasing exon skipping likelihood during co-transcriptional splicing. In general, the work showed an interesting mechanism of lariat processing. Nevertheless, there are few

major/minor points that should be considered regarding the IP-MS:

Authors overexpressed Flag-tagged Dbr1 to pull down the interactome of Dbr1. Enriched samples were separated on protein gels, and prior to mass spectrometry analysis, in gel digestion was performed. Spectral counting was used for quantitative analysis and presented AQR as a key interacting protein of Dbr1, which was validated with immunofluorescence imaging.

The p-value resulting from the statistical analysis of AQR is 0.23. Based on this value, it is challenging to conclude that AQR is a significant candidate as an interacting protein of Dbr1. Additionally, the reproducibility of results between replicates remains uncertain. While the Dbr1-AQR interaction was confirmed through western blot, it is essential to approach this result cautiously due to potential limitations in the pull-down method used by the authors. This method may co-precipitate proteins that were previously bound to DNA/RNA itself, not just proteins interacting with the bait protein. To enhance the validity of the findings, I recommend conducting additional experiments after shearing the nucleotides or DNA/RNAase treatment.

While spectral counting is considered to have some quantitative aspects, it is primarily semi-quantitative and has certain drawbacks compared to other quantification methods. For more reliable and precise quantification, I strongly recommend utilizing actual quantitative methods like label-free quantification (LFQ) or isobaric/isotopic (TMT/SILAC) labeling methods.

Minor points

I recommend using consistent terminology regarding mass spectrometry.

- Line 24: mass spectroscopy

LC-MS analysis method is not described in the method section.

Reviewer #3 (Remarks to the Author):

Buerer et al used WT and Dbr1 knockout cell lines to sequence intron lariats. Dbr1 being the debranching enzyme of the spliceosome, accumulation of intron lariats was observed upon knockout of Dbr1. The authors characterized the features of the 5' splice site, branchsite, poly-pyrimidine tract but not the 3' splice sites of the intron lariats that are debranched by Dbr1. The authors also clearly showed that Dbr1-mediated spliceosome recycling is important for efficient splicing. This a well written manuscript on an important subject. However, some of the conclusions regarding the mechanisms of action of Dbr1 need strengthening.

1. In Fig 1D, in 293T cells (the left panel), Taok2 intron 13 (red dots) are visible in both the outside (black area) and inside (blue area) the nucleus. The introns visible within the blue area may be present at the same vertical column as the nucleus but in different horizontal planes. Therefore, the conclusion that the introns are present within the nucleus may not be correct. If I am missing something which confirms that the introns are within the nucleus, it is not clear from the text. On the other hand, the total number of introns (red dots) is lower in the 293T cells (left panel) compared to the C22 cells (right panel), which is indicative of a higher level of debranching activity in the 293T cells. Therefore, this conclusion is standalone and does not need to be correlated to the sub-cellular localization of the intron.

2. Could the authors provide more background information regarding the atypical non-A branchpoint in the Introduction for a broader audience?

3. Upon Dbr1 depletion, accumulation of some introns is reduced compared to WT cells (Figure 2B). This is interesting. The authors should identify the introns whose accumulation is reduced upon Dbr1 depletion and comment on their features.

4. Figure 3A: It is not indicated whether the nuclear extract was treated with RNase A before pull-down. If not, many of the interactions reported here could be RNA-mediated. The identified interactions mediated by introns of a number of transcripts cannot be used to draw a generalized conclusion regarding protein-protein interactions.

5. In line 211, the following area should be written more clearly. "For each RBP identified in the DBR1

pull down assay, the list of introns that contained at least one RBP binding site identified by eCLIP was overlapped with the lariat abundance data derived from wildtype and DBR1 knockout samples. The lariat enrichment of the set of RBP bound introns in DBR1 knockout background relative to wildtype was compared to the intron set without a binding site.”

6. Figure 4A: The total lariats identified in C22 cells with and without publicly available eCLIP peaks should be identical or nearly identical for all RBPs for this plot to be of value. However, it appears that RNA-seq coverages for different eCLIP experiments are different. If the lariats identified in C22 line is not even covered in an eCLIP sequencing experiment for the non-AQR proteins, the conclusion that AQR is particularly important for Dbr1-mediated debranching and the statement “None of the other Dbr1 binding partners with available eCLIP data showed a similar response in lariat levels” are greatly weakened.

7. Fig 4C: The authors should identify the introns that appear in Dbr1 KO, AQR KO, and AQR eCLIP peaks for a clearer picture and a stronger conclusion.

8. Again, the pull-down assay shown in Figure 4B is not performed after RNase treatment. Many recent papers indicate that in order to work together with a common aim, two proteins do not need to strongly interact with each other. The interaction may be absent, transient, or may be too weak to be detected by pull-down assay. This does not nullify the conclusion that AQR and Dbr1 work together to debranch certain introns. However, if the authors would like to conclude that AQR and Dbr1 physically interact, the interaction must be tested in the presence of RNase A.

DBR1 Manuscript Response to Reviewers

Reviewer #1:

Comment: The authors created the DBR1 KO cells via CRISPR, and generated two (nearly) KO clones C19 and C22. Have the authors confirmed absence of off-target effects?

Response: We have performed 2x100bp whole genome sequencing on C22 and HEK293T samples in order to test for potential off-target effects of the CRISPR editing. This resulted in 711 million reads for C22 (45.9x coverage) and 723 million reads for HEK293T (46.7x coverage). Using Cas-OFFinder to detect potential off-target sites for the sgRNA we used (allowing for up to 3 mismatches and 1 DNA or RNA bulge), we found 1458 off-target sites. After read filtering, mapping and variant calling, we identified 100,495 variants which appeared in C22 but not in HEK293T. None of these variants overlapped the predicted off-target sites. More detail regarding our WGS analysis is provided in the methods.

Comment: In most experiments, results using the C22 clone were shown, but not C19. Do the C19 results closely replicate those of C22?

Response: Yes, C22 and C19 were almost indistinguishable in the functional assays performed. The C22 clone was chosen as our main model early on as its slightly lower Dbr1 levels (Fig. 1A) and slightly higher increase in lariats (Fig. 2A) relative to C19 lead us to conclude that it was a more successful knockout. The analyses that were performed on both clones (debranching assay, lariat recovery rate, intronic coverage analysis) shows that C19 replicates the C22 results. We confined the majority of our analyses to C22 as we decided to proceed with deeper sequencing of that clone. As the DBR1 deficient cell line is a unique resource and C22 had already been disseminated to other groups, we felt there was more value to characterizing this line.

Comment: Although the authors showed that DBR1 KO did not cause global gene expression shifts, there are a considerable number of genes whose expression levels are altered (Fig. 6A). Thus, in several analyses where intron coverage in sequencing was examined and compared between samples (e.g. Fig. 2B, 4C), a normalization against gene expression is needed.

Response: This is a valid concern. To remedy this issue, we have updated the analysis in Fig. 2B to normalize the intron coverage by the transcript expression level. The new figure presents the coverage of each intron as reads/TPM (transcripts per million; calculated with the Salmon transcript quantification software). Fig. 5A has also been updated with a more appropriate gene-level normalization (lariat reads/TPM). For Fig. 4C and other analyses showing lariat levels at a global scale, it should not be necessary to normalize by individual gene expression levels. We present these analyses in units of lariat reads per million mapped reads. This metric captures the proportion of a sample's reads which are lariat reads, so it is a measure of overall lariat abundance within an RNA-seq experiment that is independent of the expression level of particular genes.

Comment: In Fig. 2C, the branchpoint motif is relatively weak in 293T cells (upper

panel). Could this indicate technical noise in lariat capture experiments? Or does it reflect a depletion of Dbr1 targets due to their rapid degradation?

Response: We interpret the weak, non-canonical branchpoint motif recovered from wildtype cells as an indication that the lariats available for sequencing in these cells have already been acted upon by Dbr1. As lariats from knockout cells are predominantly those with the canonical motif, these lariats must be efficiently processed by Dbr1 in the wildtype state for them to be absent in HEK293T samples. This discrepancy leads us to conclude that Dbr1 preferentially debranches lariats with the canonical YTNAY motif, and previous research cited in our manuscript has indicated Dbr1's preference for "A" branchpoints (Nam et al. 1994).

Comment: What are the interpretations of the sequence preferences in 5' splice sites in Fig. 2D?

Response: Dbr1 is already known to preferentially debranch 'A' branchpoints. We hypothesize that the observed differences in motif reflect Dbr1's substrate preference. In other words, the enrichment of 'G's in knockout, suggests Dbr1 prefers 'G' residues at positions +3 and +5 relative to the 5' splice site.

Comment: The model that AQR recruits DBR1 is interesting. More supporting evidence will be preferred, for example, the authors could mutate the AQR binding motifs and test for DBR1 recruitment.

Response: We tried to develop a quantitative assay for lariat PCR, however in our hands, the transient nature of the DBR1 interaction and the extremely low concentration of lariats precluded PCR based quantitative experiments on single substrates.

Comment: Fig. 5C, although a low p value was shown, the effect size is very small. The authors should rule out the possible contribution of intron retention in this analysis.

Response: We apologize if the experimental design was not clear. We clarify that the experiment performed involved lariat PCR, i.e. PCR using an inverted primer strategy (a reverse primer upstream of a forward primer) that is only able to amplify circular species. Thus, the sequenced products could only originate from introns which have already undergone splicing, so the presence or absence of intron retention would not interfere with the result.

Comment: Fig. 5E, the RBM22 results appear to show a smaller change in editing rates (C22 vs 293T) compared to PPIE. What are the authors interpretations? Does it reflect a limitation of the assay that ADAR activity may differ in different fusion contexts?

Response: Yes, I think your idea is a possible explanation. The difference could also be explained by access to substrate. ADAR fusions may be excluded by their endogenous counterparts from functional complexes to a greater or lesser extent. Such a difference could explain the lower editing rates observed in the RBM22 experiment. While this may limit the sensitivity of the assay, both cell lines are transfected with the same fusion protein, so the consistent, significant increase in editing rates observed in C22 still supports the longer association times of these fusions with the lariat substrates.

Reviewer #2:

Comment: The p-value resulting from the statistical analysis of AQR is 0.23. Based on this value, it is challenging to conclude that AQR is a significant candidate as an interacting protein of Dbr1. Additionally, the reproducibility of results between replicates remains uncertain. While the Dbr1-AQR interaction was confirmed through western blot, it is essential to approach this result cautiously due to potential limitations in the pull-down method used by the authors. This method may co-precipitate proteins that were previously bound to DNA/RNA itself, not just proteins interacting with the bait protein. To enhance the validity of the findings, I recommend conducting additional experiments after shearing the nucleotides or DNA/RNAase treatment.

Response: We agree that this issue requires further attention. We measured an interaction between Dbr1 and AQR using co-IP/MS. We validated this by repeating the co-IP, with an additional control of the inverse IP (AQR pulldown) and analyzing by western blot. We demonstrated that AQR and Dbr1 co-localize with immunofluorescence microscopy, and that knocking down AQR elevated lariat levels (Fig. 4C). Furthermore, a recent publication from a collaborator (DOI: 10.1016/j.molcel.2023.06.011) demonstrates that Dbr1 is tethered to the intron binding complex (which contains AQR) by the TTDN1 protein. Given this new information, we felt that we cannot rule out an indirect interaction. Even if we demonstrate a pulldown that is robust to nuclease treatment, the interaction could still be occurring through any protein in the intron binding complex.

To address the reviewers concerns, we have added discussion of these potential complications to Discussion paragraph #4. We clarify that the IP was performed without nuclease treatment, and that the interactions could be indirect, mediated by either nucleic acids or protein in Results section #3. We do demonstrate a functional enhancement of debranching which is dependent on the proximity of an AQR site to the branch site.

Comment: While spectral counting is considered to have some quantitative aspects, it is primarily semi-quantitative and has certain drawbacks compared to other quantification methods. For more reliable and precise quantification, I strongly recommend utilizing actual quantitative methods like label-free quantification (LFQ) or isobaric/isotopic (TMT/SILAC) labeling methods.

Response: We modified the text to refer to our co-IP mass spectrometry experiments as "semi-quantitative" in the results and methods sections. We try to make clear that this technique was used as a discovery tool and not a method to quantify the binding partners. We appreciate the reviewer's suggestion of more preferred quantitative mass spectrometry approaches, and we will consider these in future experiments.

Reviewer #3:

Comment: In Fig 1D, in 293T cells (the left panel), Taok2 intron 13 (red dots) are visible in both the outside (black area) and inside (blue area) the nucleus. The introns visible within the blue area may be present at the same vertical column as the nucleus but in different horizontal planes. Therefore, the conclusion that the introns are present within the nucleus may not be correct. If I am missing something which confirms that the

introns are within the nucleus, it is not clear from the text. On the other hand, the total number of introns (red dots) is lower in the 293T cells (left panel) compared to the C22 cells (right panel), which is indicative of a higher level of debranching activity in the 293T cells. Therefore, this conclusion is standalone and does not need to be correlated to the sub-cellular localization of the intron.

Response: This is a good critique to make but the depth of field for this microscope makes this type of contamination very unlikely. Using a 100x 1.45NA objective, the depth of field is quite small. An online calculator estimates the depth of field of this objective as approximately 0.5 um which is about 1/10 the diameter of a typical nucleus. So when focused on a nucleus, you'll image relatively little above or below the focal plane (See: <https://www.translatorscafe.com/unit-converter/en-US/calculator/depth-of-field/>

with the following settings: 550 nm light, synthetic immersion oil, NA 1.45, 100x, 10 um pixels (estimated); Depth of field = 0.5 um). We have modified the methods to include this data.

Even if it were the case that the nuclear column may contain some portion of the cytoplasm, the dots outside the outline of the nucleus are unambiguously cytoplasmic. The measure may undercount the cytoplasmic fraction but this would make it harder, not easier, to detect a difference. As these calculations indicate very little cytoplasmic contamination and the same approach was used in the DBR1 wildtype and depleted cells we believe this analysis is valid.

Comment: Could the authors provide more background information regarding the atypical non-A branchpoint in the Introduction for a broader audience?

Response: This is a good suggestion. Probably the best introduction is Konarska and Query's systematic analysis of the spliceosome's ability to utilize non-canonical branchsites (10.1016/j.molcel.2009.03.012). We added this to the introduction.

Comment: Upon Dbr1 depletion, accumulation of some introns is reduced compared to WT cells (Figure 2B). This is interesting. The authors should identify the introns whose accumulation is reduced upon Dbr1 depletion and comment on their features.

Response: This is an interesting observation that we did not think to test. There are not many of these introns that appear to decrease with DBR1 knockdown. We analyzed this set of introns for features such as sequence motifs, branchpoint nucleotide, intron length, etc. but none of these showed differences when compared to introns that increase upon Dbr1 depletion. This result caused us to suspect that this decrease may be a gene-level phenomenon caused by decreased transcription in the KO relative to WT, the introns did not cluster in specific genes. Further analysis of the coverage levels for this set of reduced introns indicates that the decrease is due to the very high accumulation of these introns in WT, not a particular low transcription relative to other introns in the KO. Compared to introns which increase in normalized coverage upon Dbr1 depletion, introns that decrease have similar coverage levels within the KO samples (Fig. S2A). The difference comes in the WT, where these introns have extraordinarily high coverage values relative to the other introns (Fig. S2B). Thus, the KO vs WT difference observed for these introns appears to be a fluctuation in stability or

perhaps a spurious amplification noise that is occurring in the WT sample rather than a consequence of the KO.

Comment: Figure 3A: It is not indicated whether the nuclear extract was treated with RNase A before pull-down. If not, many of the interactions reported here could be RNA-mediated. The identified interactions mediated by introns of a number of transcripts cannot be used to draw a generalized conclusion regarding protein-protein interactions.

Response: This is a valid concern and we have altered the text for clarification. We clarify that the IP was performed without nuclease treatment, and that the interactions could be indirect, mediated by either nucleic acids or protein in Results section #3 and Discussion paragraph #4. We do demonstrate a functional enhancement of debranching which is dependent on the proximity of an AQR site.

Comment: In line 211, the following area should be written more clearly. “For each RBP identified in the DBR1 pull down assay, the list of introns that contained at least one RBP binding site identified by eCLIP was overlapped with the lariat abundance data derived from wildtype and DBR1 knockout samples. The lariat enrichment of the set of RBP bound introns in DBR1 knockout background relative to wildtype was compared to the intron set without a binding site.”

Response: I'm sorry for the lack of clarity. We have rewritten that section as follows: For each RBP identified in the DBR1 pull down assay, the set of introns that contained at least one RBP binding site (i.e. eCLIP enrichment) was identified. As a control, the same number of introns was sampled from the set of introns with no reported eCLIP sites for any of the analyzed RBPs. The normalized lariat levels in wildtype and DBR1 knockout samples were then calculated for each intron set, and the lariat enrichment of the DBR1 knockout relative to wildtype was compared between the bound and non-bound intron sets.

Comment: Figure 4A: The total lariats identified in C22 cells with and without publicly available eCLIP peaks should be identical or nearly identical for all RBPs for this plot to be of value. However, it appears that RNA-seq coverages for different eCLIP experiments are different. If the lariats identified in C22 line is not even covered in an eCLIP sequencing experiment for the non-AQR proteins, the conclusion that AQR is particularly important for Dbr1-mediated debranching and the statement “None of the other Dbr1 binding partners with available eCLIP data showed a similar response in lariat levels” are greatly weakened.

Response: The experiment measures the effect of RBP binding on an intron's sensitivity to Dbr1. Your concern about coverage is correct, and we have updated the analysis by normalizing for expression levels in the data presented in this panel. The updated panel presents the lariat levels of each set of bound introns normalized by the expression levels of the genes containing those introns (reads/transcripts per million). While it is true that there are many more introns reported to be bound by AQR than the other RBPs which interact with Dbr1, we do not believe this weakens the analysis. The comparison we are attempting to make is between the bound and non-bound intron sets for each RBP. Since the non-bound intron set is matched in size to the bound intron set, this comparison should provide a valid indication of the responsiveness of that particular

set of introns to DBR1 knockout. This does not preclude the potential contribution of other RBPs to lariat recycling or the existence of introns that are able to be efficiently degraded without the presence of these RBP binding sites. Rather, we interpret the accentuated increase in lariats upon DBR1 knockout for the AQR-bound introns as simply an initial indication that some type of Dbr1-AQR interaction could be mediating lariat processing for that subset of introns. Indeed, the updated analysis performed with gene level expression normalization as described above reveals a similar response for RBM22 and SRSF7. Nevertheless, the follow-up analyses presented in this figure (particularly the lariat increases observed upon AQR knockdown) support the conclusion that AQR is involved in promoting lariat degradation in at least a subset of introns.

Comment: Fig 4C: The authors should identify the introns that appear in Dbr1 KO, AQR KO, and AQR eCLIP peaks for a clearer picture and a stronger conclusion.

Response: We have added analysis of the introns that appear as lariats in the DBR1 KO and AQR KD and their overlap with annotated AQR eCLIP sites in Supplemental Figure 4. There is good overlap between the introns recovered from the DBR1 KO and AQR KD (Supplemental Fig. 4A). In addition, comparing the introns with recovered lariats in Dbr1 KO and AQR KD to the AQR binding sites shows that the presence of an AQR site in an intron increased the likelihood of recovering a lariat from that intron in both the Dbr1 KO and AQR KD experiments (Supplemental Fig. 4B/C). This increase in lariat recovery for AQR-bound introns supports the hypothesis that AQR sites promote the efficient degradation of the lariats containing them.

Comment: Again, the pull-down assay shown in Figure 4B is not performed after RNase treatment. Many recent papers indicate that in order to work together with a common aim, two proteins do not need to strongly interact with each other. The interaction may be absent, transient, or may be too weak to be detected by pull-down assay. This does not nullify the conclusion that AQR and Dbr1 work together to debranch certain introns. However, if the authors would like to conclude that AQR and Dbr1 physically interact, the interaction must be tested in the presence of RNase A.

Response: This is a good point. We have updated the text and figures to clarify that the proposed Dbr1-AQR interaction may be indirect, either protein- or RNA-mediated (Results section #3, Discussion paragraph #4).

REVIEWERS' COMMENTS

Reviewer #1 (Remarks to the Author):

Thank you for addressing my concerns.

Reviewer #1 (Remarks on code availability):

The code is usable, and will be valuable to the community.

Reviewer #2 (Remarks to the Author):

The manuscript was carefully revised. All of my questions were appropriately met and answered. I do not have any further questions and recommend acceptance of the manuscript in its present form.

Reviewer #3 (Remarks to the Author):

The authors have adequately addressed my concerns.